# A Risk Score for Predicting Long-Term Mortality Following Off-Pump Coronary Artery Bypass Grafting

**DOI:** 10.3390/jcm10143032

**Published:** 2021-07-07

**Authors:** Tomasz Kamil Urbanowicz, Michał Michalak, Aleksandra Gąsecka, Anna Olasińska-Wiśniewska, Bartłomiej Perek, Michał Rodzki, Michał Bociański, Marek Jemielity

**Affiliations:** 1Department of Cardiac Surgery and Transplantology, Poznan University of Medical Sciences, 61-701 Poznan, Poland; anna.olasinska@poczta.onet.pl (A.O.-W.); bperek@ump.edu.pl (B.P.); michal.rodzki@skpp.edu.pl (M.R.); michal.bocianski@wp.pl (M.B.); mjemielity@poczta.onet.pl (M.J.); 2Department of Computer Science and Statistics, Poznan University of Medical Sciences, 61-701 Poznan, Poland; michal@ump.edu.pl; 31st Chair and Department of Cardiology, Medical University of Warsaw, 02-091 Warsaw, Poland; aleksandra.gasecka@wum.edu.pl

**Keywords:** coronary artery bypass grafting, off-pump, multivessel coronary artery disease, risk stratification, prediction, long-term mortality, neutrophil-to-lymphocyte ratio

## Abstract

Background: Off-pump coronary artery bypass grafting (OPCAB) comprises 15–30% of all bypass grafting surgeries. The currently available perioperative scores such as Euroscore and STS score do not specifically predict long-term mortality after off-pump procedures. The neutrophil-to-lymphocyte ratio (NLR) is one of the new, easily accessible markers of inflammation with proven predictive value in cardiovascular diseases. We aimed to develop the first risk score for long-term mortality after OPCAB and to determine if the perioperative value of NLR predicts long-term mortality in OPCAB patients. Methods: In total, 440 consecutive patients with multivessel stable coronary artery disease undergoing OPCAB were recruited. Differential leukocyte counts were obtained by a routine hematology analyzer. Data regarding mortality during a median follow-up time of 5.3 years were obtained from the Polish National Health Service database. An independent population of 242 patients served as a validation cohort. Results: All-cause mortality was influenced by different clinical risk factors. In multivariate regression analysis, chronic obstructive pulmonary disease, stroke history, post-operative NLR and LVEF were independent predictors of mortality. Combing all independent predictors predicted long-term all-cause mortality with 68.5% sensitivity and 71.5% specificity (AUC = 0.704, *p* < 0.001). After weighing these variables according to their estimates in a multivariate regression model, we developed a score to predict mortality in patients undergoing OPCAB (PREDICT-OPCAB Score, ranging from 0 to 10). Patients with a high score were at higher risk of mortality within the median 5.3 years of follow-up (score 0–3: 8.3%; 4–6: 27.0%; 7–10: 40.0%; *p* < 0.001 for score 0–3 vs. 4–6 and 7–10). This association was confirmed in the validation cohort. Conclusions: We developed and validated the first simplified risk score to predict mortality following OPCAB based on easily accessible clinical factors. This risk score can be used when obtaining a patient’s informed consent and as an aid in determining treatment.

## 1. Introduction

Coronary artery bypass grafting (CABG) surgery remains the gold standard to treat multivessel coronary artery disease (MVD) with high lesion complexity due to its survival advantage over percutaneous coronary interventions (PCI) [1]. Surgical revascularization can be performed with (on-pump) or without (off-pump) cardiopulmonary bypass application, with satisfactory outcomes [2]. Currently, off-pump coronary artery bypass grafting (OPCAB) comprises 15–30% of all CABG cases, based on different national registries [2,3]. Whereas the short-term patency of grafts is excellent [4], the risk of long-term graft failure ranges from 10 to 15% for arterial grafts and from 20 to 40% for venous grafts up to 9 years following surgery [5].

Currently, perioperative risk scores such as Euroscore II or STS score are available and routinely applied before surgery [6,7]. Simplified risk scores have been developed to predict short- and long-term mortality after on-pump CABG [8,9]. However, no risk score currently exists to predict long-term mortality after off-pump procedures. The development of such score, specifically related to perioperative factors, including preoperative data combined with intra- and postoperative parameters, would enable clinicians to identify the subgroup of patients with poor prognosis and subsequently focus on modifiable risk factors and treatment intensification.

Chronic and acute inflammatory reactions are involved in every stage of atherosclerosis, from atherosclerotic plaque formation, via steady progression, to acute coronary syndromes (ACS) [10]. There is also a protracted postoperative activation of inflammation persisting several days after CABG [11]. The effects of on-pump CABG on inflammatory system activation and endothelial dysfunction have been well described [12,13]. Off-pump CABG was found to reduce the systemic inflammatory response, although recent studies suggest that, with the use of biocompatible extracorporeal circuits, the inflammatory response triggered by on-pump and off-pump CABG might be similar [14,15]. Due to the differences in the pathophysiological response to on- and off-pump CABG, we hypothesized that the risk factors associated with an adverse prognosis might differ as well.

The neutrophil-to-lymphocyte ratio (NLR) is one of the new, easily accessible markers of inflammation with proven predictive value in cardiovascular diseases. NLR is calculated by dividing the number of neutrophils by the number of lymphocytes. Under stress conditions, such as open-heart surgery, the number of neutrophils increases, while the number of lymphocytes decreases. The NLR combines changes in both parameters, making it more sensitive than either value alone [14,15]. NLR was an independent predictor of outcomes following PCI among acute coronary syndrome (ACS) subjects [16,17]. Since OPCAB triggers an inflammatory response, we hypothesized that NLR might also be a useful parameter that affects prognosis in this setting. However, the association between NLR and OPCABG outcomes have not been estimated so far. Therefore, the aim of our study was to (i) to determine if the perioperative value of NLR predicts long-term mortality after OPCAB and (ii) to develop the first risk score for long-term mortality.

## 2. Patients and Methods

### 2.1. Study Population

We conducted a single-center registry-based retrospective analysis of patients undergoing OPCAB at an experienced academic center in Poland. The trial was conducted in accordance with the principles of Good Clinical Practice and the Declaration of Helsinki. The study protocol was approved by the Ethics Committee of Medical University of Poznan (approval number: 55/2). All participants provided written informed consent.

The study population comprised 440 consecutive patients who underwent OPCAB surgeries between January 2014 and December 2017. The mean follow-up was 63.7 ± 13.5 months (5.3 years), with a range of 12.0 to 80.9 months (1.0–6.7 years). All patients had MVD and were qualified for elective CABG by the Heart Team [18]. We excluded all patients with acute nonST-elevation (NSTEMI) and ST-elevation (STEMI) myocardial infarction, concomitant valve disease requiring surgical valve replacement, and end-stage kidney disease requiring hemodialysis. In addition, because the NLR is affected by hematological proliferative diseases, active or chronic inflammatory or autoimmune diseases, and steroid therapy, patients with these conditions were excluded as well. Finally, since NLR was the primary study endpoint, we excluded patients with unavailable complete differential blood count pre-operatively.

We used standardized definitions to collect clinical information from electronic medical records, including patient demographics, comorbidities, laboratory data, pharmacotherapy, echocardiographic parameters, procedural details, and in-hospital outcomes.

Pre-operative NLR was determined on day 1 before surgery (baseline value). Post-operative NLR was determined on a median of 7 days after surgery (at hospital discharge). NLR is calculated as the number of neutrophils divided by the number of lymphocytes. The numbers of neutrophils and lymphocytes for NLR determination were obtained by a routine hematology analyzer (Sysmex Europe GmbH, Norderstedt Germany).

Other laboratory data including myocardial injury markers were obtained 1 day before and 1 day after surgery. Transthoracic echocardiography was performed within 3 days prior to surgery and at hospital discharge. Data regarding long-term mortality were obtained from the Polish National Health Service database.

### 2.2. Surgery

All surgeries were performed by one team of 4 experienced surgeons. The number of off-pump procedures performed by every surgeon prior the study was over 300. The OPCAB procedure is a standard technique in our institution. In case of either hemodynamic instability, especially in STEMI/NSTEMI patients or during intraoperative heart elevation, the conversion into cardiopulmonary bypass is performed. In the presented study, none of the patients were converted from off-pump to on-pump surgery.

The procedures were performed from full median sternotomy on the beating heart. The mammary arteries (left or both), radial arteries, and segments of the saphenous vein for venous aortocoronary grafts were harvested. Intravenous heparin was administered to achieve the desired activated clotting time (ACT) of 420 ± 40 s. Distal anastomoses to the epicardial portions of the stenotic coronary arteries were performed with local immobilization by means of the Octopus III (Medtronic, Minneapolis, MN, USA) and deep pericardial stitch. The anastomoses were performed by running a 7-0 monofilament suture after artery longitudinal incision was followed by intraluminal shunt application. The left mammary artery and venous grafts were used in the performed analysis, excluding the right mammary and radial artery.

### 2.3. Outcomes

The primary outcome was the predictive value of the constructed score for long-term all-cause mortality in patients with stable complex coronary artery disease referred for OPCAB procedure.

### 2.4. Statistical Analysis

Continuous variables are reported as mean ± standard deviation (SD) or median with interquartile range and were compared using an unpaired t-test or Mann–Whitney U test, depending on the normality of the distribution. Categorical variables ware reported as frequencies and percentages and were compared using a test for proportions. The cut-offs values of the analyzed predictors that discriminated between patients with and without mortality endpoint were calculated using a receiver operating characteristic (ROC) curve. The significance of the survival curves was checked by the Log-rank test. The Cox’s proportional hazards model was used to analyze potential risk predictors for long-term overall mortality. The analysis was performed both as univariate and as multivariate, adjusting the estimated coefficients to the coexistence of other predictors. The multivariate analysis was performed with a stepwise, backward selection procedure. The obtained results are presented as hazard ratios (HR) and their 95% confidence intervals (CI). The continuous parameters were transformed into binary parameters (using ROC analysis) to unify data type. This was necessary to compare the effect sizes of the obtained HR.

A multivariate Cox regression model results was used to calculate the PREDICT-OPCAB score. According to the effect size of HR for relevant clinical predictors, the score was developed using the following rule: for obtained HR ≤ 1.5, 1 pt, for 1.5 < HR ≤ 2.5, 2 pts, and for HR > 2.5, 3 pts. The PREDICT-OPCAB score for each patient was calculated as the sum of points. The long-term overall mortality was compared between different score ranges by Kaplan–Meier curves. The significance of the survival curves was checked by the Log-rank test. Next, the PREDICT-OPCAB score was validated by employing an independent cohort of 242 patients who underwent OPCAB surgeries in 2018. All tests were considered significant at *p* < 0.05. The analysis was performed by MedCalc^®^ Statistical Software version 19.6 (MedCalc Software Ltd., Ostend, Belgium). All patients included in the study met the inclusion criteria.

## 3. Results

### 3.1. Baseline Characteristics

In the total study population, the mean age was 64 ± 8 years, and 382 patients were male (86%). Comorbidities included hypertension in 340 patients (77%), diabetes mellitus in 150 (34%), hypercholesterolemia in 243 (55%), chronic obstructive pulmonary disease in 44 (10%), and chronic kidney disease defined as glomerular filtration rate ≤60 mL/min/1.63 m^2^ according to the Cockcroft–Gault equation in 33 (8%). There were 101 (23%) patients operated due to left main disease and 339 (77%) referred for surgery due to three-vessel disease. The mean surgery (skin-to-skin) time was 2.4 ± 0.4 h, and the mean number of performed anastomoses was 2.3 ± 0.2. The majority of the performed grafts included composition of the left mammary artery (LIMA) and venous grafts 420 (95%), followed by LIMA with left radial artery (LRA) 14 (3%) and both mammary arteries 7 (2%). None of the surgeries were performed as a redo surgery. The rethoracotomy incidence was 10 (2%), with mean ICU stay of 24 ± 8 h. There were only 2 cases (0.5%) of postoperative stroke and 2 cases (0.5%) of respiratory insufficiency. The study was based on all-cause mortality, excluding the incidence of postoperative coronary artery angiography throughout the period.

### 3.2. Risk Factors of Long-Term Mortality

There were no intraoperative deaths. The 30-day mortality rate was 1% (four patients). The cumulative survival rate during the median 5.3-year follow-up period was 87%. The clinical characteristics of patients with and without all-cause mortality endpoint are shown in Table 1. Patients who died were older (*p* = 0.0351), had more often chronic obstructive pulmonary disease (COPD; *p* < 0.0001), and a history of stroke (*p* = 0.0261) and peripheral artery disease (*p* = 0.0473). These patients also had a higher NLR both pre- and post-operatively (*p* = 0.0256; *p* = 0.0173), a larger left ventricle end-diastolic diameter (LVEDD) pre- and post-operatively (0.0049; 0.0017), and a lower left ventricle ejection fraction (LVEF) pre- and post-operatively (*p* = 0.0012; *p* = 0.0010).

Table 2 presents the Cox regression model to determine which clinical parameters were predictors of mortality after OPCAB surgery. In univariate analysis, left main disease (HR = 1.98, 95% CI 1.12–3.49), stroke history (HR 2.40, 95% CI 1.21–4.76), postoperative NLR (HR = 1.13, 95% CI 1.05–1.21), pre- and postoperative left LVEDD (HR 1.05, 95% CI 1.01–1.09 and HR = 1.06, 95% CI 1.02–1.10, respectively), and pre- and postoperative LVEF (HR = 0.95, 95% CI 0.93–0.97 and HR = 0.96, 95% CI 0.93–0.98, respectively) predicted long-term all-cause mortality.

### 3.3. Predictive Value of NLR for Long-Term Mortality

ROC curve analysis (Figure 1A) showed that the post-operative NLR discriminated between patients with and without the mortality endpoint (AUC = 0.598, *p* = 0.014). However, regarding the relatively poor specificity of NLR for post-operative long-term mortality, we conducted a multivariable Cox regression model to identify other independent predictors of mortality after CABG surgery and subsequently improve risk stratification.

Table 3 presents the multivariable Cox regression model that identified four independent predictors of long-term mortality after CABG surgery. The cut-off for post-operative NLR and LVEF values predictive for mortality were (i) NLR ≥ 2.42 (77.2% sensitivity, 44.0% specificity) and (ii) LVEF ≤ 45% (43.9% sensitivity, 84.0% specificity). COPD, stroke history, postoperative NLR ≥ 2.42, and post-operative LVEF ≤ 45% remained significant predictors of long-term all-cause mortality, whereas other parameters predictive in univariate analysis were no longer significant.

Combing all independent predictors of all-cause mortality (COPD, history of stroke, post-operative NLR and LVEF) improved the long-term all-cause mortality prediction (AUC = 0.704, *p* < 0.001), allowing to predict mortality with 68.5% sensitivity and 71.5% specificity (Figure 1B).

### 3.4. Development and Performance of the Risk Score

According to effect size of HR of the relevant clinical predictors of all-cause mortality in multivariate Cox regression analysis, a weighed score was developed. In detail, COPD was weighted by factor 2, stroke history by factor 3, NLR ≥ 2.42 by factor 2, and LVEF ≤ 45 by factor 3 (hazard ratios 2.19, 2.67, 1.96, and 2.64, respectively). Thus, a score ranging from 0 to 10 was developed (Table 3).

The study population was divided into three subgroups, depending on the PREDICT-OPCAB score: 0–3 points, 4–6 points, and 7–10 points. Figure 2 shows the Kaplan–Meier curves according to the PREDICT-OPCAB score for the study population. The log-rank test showed that survival rates significantly differed between the subgroups (*p* < 0.0001).

Table 4 shows the mortality rates for the study population (upper panel) according to the PREDICT-OPCAB score. There was an increase in long-term mortality with a higher PREDICT-OPCAB score, i.e., 8.3% for patients with a score of 0–3, 27.0% for patients with a score of 4–6, and 40.0% for patients with a score of 7–10 (*p* < 0.0001 for 0–3 vs. 4–6 and for 0–3 vs. 7–10 pts). There was no significant difference in mortality between patients who had a score of 4–6 points vs. 7–10 points (*p* = 0.1100).

### 3.5. Validation of the Risk Score in an Independent Cohort

For validation of the PREDICT-OPCAB score, we recruited 242 consecutive patients according to the same inclusion criteria as those used for the study population undergoing CABG at our department between January and December 2018. The median follow-up time was 2.6 years (0.48–3.07). The clinical characteristics of patients with and without the all-cause mortality endpoint during the median of 2.6 years follow-up in the validation group are shown in Table 5. In the validation cohort, the mean age was of 69 ± 9 years, and 198 were male (82%) mean. Comorbidities included arterial hypertension in 187 patients (77%), diabetes mellitus in 87 (35%), hypercholesterolemia in 152 (63%), chronic obstructive disease in 19 (8%), and chronic kidney disease in 22 (9%). There were 46 (19%) patients operated due to left main disease, and 196 (81%) referred for surgery due to three-vessel disease. The mean surgery (skin-to-skin) time was 2.4 ± 0.3 h, and the mean number of performed anastomoses was 2.25 ± 0.2.

There were no intraoperative deaths. The 30-day mortality rate was 0.5% (one patient). The cumulative survival rate during the median 2.6 years follow-up period was 89%. The clinical characteristics of patients with and without the all-cause mortality endpoint in the validation cohort are shown in Table 5. Patients who died had more often COPD (*p* < 0.0001), a history of stroke (*p* < 0.0001), and peripheral artery disease (*p* = 0.0005). These patients also had higher post-operative NLR value and troponin concentration (*p* = 0.0003; *p* = 0.0113), larger post-operative LVEDD (*p* = 0.0205), lower pre- and post-operative LVEF (*p* = 0.0030; *p* < 0.0001), and longer in-hospital length of stay (*p* = 0.0030).

Figure 3 shows the Kaplan–Meier curves according to the PREDICT-OPCAB score for the validation cohort. The log-rank test showed that the survival rates significantly differed between the subgroups (*p* < 0.0001). Similar to the study population, there was an increase in long-term mortality with a higher PREDICT-OPCAB score (Table 4), i.e., 3.9% in patients with a score of 0–3, 40.7% in patients with a score of 4–6, and 77.8% in patients with a score of 7–10 (*p* < 0.0001 for 0–3 vs. 4–6 and for 0–3 vs. 7–10 pts). There was no significant difference in mortality between patients who had a score of 4–6 points vs. one of 7–10 points (*p* = 0.0838).

## 4. Discussion

The most important result of the study is the definition of a new score that predicts long-term mortality in patients undergoing OPCABG, including the NLR as a significant risk factor.

In this study, we used a Cox proportional hazards model that identified the independent risk factors for long-term mortality up to 7 years after OPCAB surgery for patients undergoing surgical revascularization due to complex stable coronary artery disease excluding acute NSTEMI/STEMI. The main finding of this study is the creation of constitutive factors, considering that the postoperative NLR value measured at hospital discharge is an independent predictor of long-term mortality after OPCAB. Since NLR specificity was not satisfactory, we incorporated NLR and other clinical parameters to develop and validate the first simplified risk score to predict mortality following OPCAB.

We found that COPD, stroke history, post-operative LVEF ≤ 45%, and post-operative NLR ≥ 2.42 were independent risk factors of mortality. In a recent meta-analysis of eight clinical trials, patients with COPD undergoing CABG were at higher risk of post-operative complications, including pneumonia, respiratory failure, stroke, renal failure, and wound infection, although the short-term mortality rate was comparable between patients with and without COPD [19]. In addition, although CABG is known to have long-term beneficial effects in patients with COPD [20], more severe COPD was associated with worse clinical outcomes after OPCAB surgery [21]. Although we did not analyze the short-term outcomes in the subpopulation of COPD patients and did not stratify the results according to COPD severity, our results indicate that COPD is associated with a two-fold higher long-term mortality.

Whereas numerous studies focused on the risk of post-operative stroke after CABG [22,23], there are few data regarding the association between the pre-operative cerebrovascular episodes and post-operative mortality. In one study comprising a large cohort of 62,104 patients who underwent CABG, prior stroke increased two-fold both the risk of post-operative stroke and mortality [24]. In our study, prior stroke was associated with a three-fold increase in long-term mortality, indicating that stroke history should be taken into account when obtaining a patient’s informed consent for determining the treatment option, especially, a higher rate of stroke after CABG compared to PCI, as shown in a recent meta-analysis [22].

Reduced LVEF is an established risk factor for perioperative complications and mortality after CABG [8,25]. Although recent studies indicate that off-pump CABG may be associated with benefits compared to on-pump CABG in reduced LVEF patients [26,27], all-cause and cardiovascular mortality was increased in this subset of patients [27]. In our cohort, LVEF ≤ 45% increased the risk of long-term mortality three-fold.

In addition to the classical risk factors, we demonstrated that a higher NLR increased the risk of all-cause mortality after OPCAB. The NLR can be calculated based on differential leukocyte counts by a routine hematology analyzer, which makes it an inexpensive and easily accessible indicator of systemic inflammation in daily clinical practice. The NLR was showed to be of additional value for the short-term risk stratification of patients with myocardial infarction undergoing PCI [28,29], coronary heart disease patients undergoing non-cardiac surgery [30], and those undergoing on-pump CABG [31]. To the best of our knowledge, this is the first study showing the predictive value of NLR for long-term mortality after OPCAB. Post-operative NLR ≥ 2.42 increased the mortality risk two-fold in our cohort, suggesting that an increased systemic inflammatory response after OPCAB deteriorates the prognosis besides classical risk factors, even despite the elimination of the pro-inflammatory effect of the cardiopulmonary bypass. Noteworthy, the sensitivity of the post-operative NLR measurement and that of the created score were comparable, but the score had higher specificity and better discriminative power (Figure 1). Since the developed score consists of clinical values which are easily accessible, we propose to use this score rather than only the NLR measurement to predict mortality after OPCAB. Although the predictive performance of NLR is mediocre (Figure 1), the NLR is the only parameter which reflects an ongoing inflammation and the only potentially modifiable parameter out of four clinical variables included in the score. Multiple strategies to decrease atherosclerosis-related inflammation have been tested in clinical trials, including a monoclonal antibody targeting interleukin-1β (canakinumab) [32], colchicine [33], or methotrexate [34]. However, at present, none of these treatments is recommended in clinical routine due to lack of data regarding the efficacy–safety balance and cost-effectiveness. Hence, the only action to modify the postoperative NLR is to minimize the perioperative stress, for example, by avoiding excessive manipulations during the operation and the use of biocompatible extracorporeal circuits during OPCAB.

Gender, age, and body mass index (BMI) were not independent predictors of long-term mortality in this study, although the female gender has been reported to be associated with a higher risk of short-term and long-term mortality after CABG [35,36]. Age was associated with survival after CABG in previous studies [8,37]. Likely, the lack of prognostic value of age in our study was due to the fact that our study cohort was relatively young (mean age 64 ± 8 years) and homogenous. Finally, BMI was not associated with mortality in our study population, although overweight and obese patients were shown to have lower mortality after cardiac surgery compared with normal-weight and underweight patients (“obesity paradox”) [38].

## 5. Limitations

The main limitation of our study is that our data were obtained from a single center, which might limit the generalizability of the results. Second, this study was developed based on the data of patients who underwent OPCAB a median of 5.3-years ago, referred for surgery, excluding NSTEMI/STEMI patients. The quality of OPCAB surgery, the implementation of biocompatible extracorporeal circuits, and follow-up care have improved over time, which might be associated with better outcomes following OPCAB. Third, although many cardiovascular and non-cardiovascular risk factors (COPD, stroke history, chronic kidney disease, diabetes mellitus) were considered, other risk factors for long-term mortality (for example cancer, smoking status) were not available. Also, some risk factors (e.g., body mass index) may change over time, but only the baseline values were available. Addition of other established risk factors to the model might improve the prognostic value for mortality. Fourth, factors such as active endocarditis, critical preoperative state, or recent myocardial infarction, which are a part of the EuroScore, were included in exclusion criteria in this study. Hence, our score only applies to stable patients undergoing elective OPCAB. Fifth, the developed score predicts all-cause mortality but not cardiovascular mortality, nonfatal myocardial infarction, stroke, and graft failure, which are of great interest as well. However, since this was a registry-based study with no independent event adjudication committee, only data regarding long-term mortality, obtained from the Polish National Health Service database, were considered a reliable endpoint. Sixth, since our score includes two post-operative variables (LVEF and NLR), it is intended to predict the long-term mortality at hospital discharge and not to give predictions prior to surgery. Finally, the vast majority of the performed grafts included composition of the left mammary artery and venous grafts. Altogether, although our risk score should be further validated in other studies, it summarizes the most recent and complete data on post-OPCAB long-term mortality currently available.

## 6. Conclusions

We developed and validated the first simplified risk score to predict mortality following OPCAB based on easily accessible clinical factors. We found that four factors including stroke history, postoperative left ventricle ejection fraction, and postoperative NLR measured at hospital discharge combined with COPD may generate a new predictive score for complex stable coronary artery disease. This risk score can be used to identify especially vulnerable patients at hospital discharge, enabling the intensification of check-up visits and treatment in these patients. In future studies, it would be of great interest to evaluate the predictive value of NLR and to validate our score also in patients undergoing on-pump CABG.

## Figures and Tables

**Figure 1 jcm-10-03032-f001:**
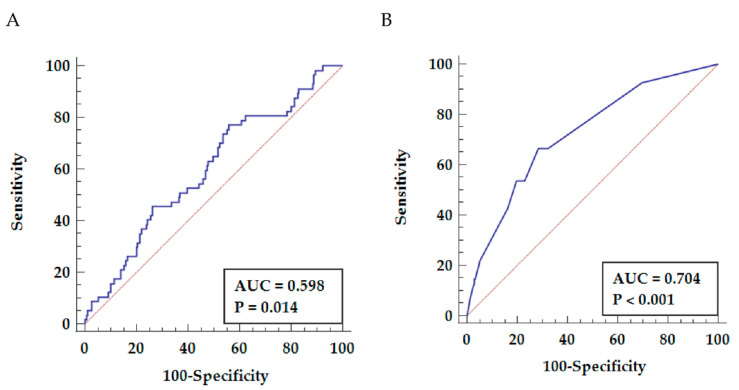
ROC curves predictive for all-cause mortality. (**A**) Predictive value of NLR (77.2% sensitivity, 44.0% specificity). (**B**) Predictive value of the PREDICT-OPCAB score (68.5% sensitivity and 71.5% specificity).

**Figure 2 jcm-10-03032-f002:**
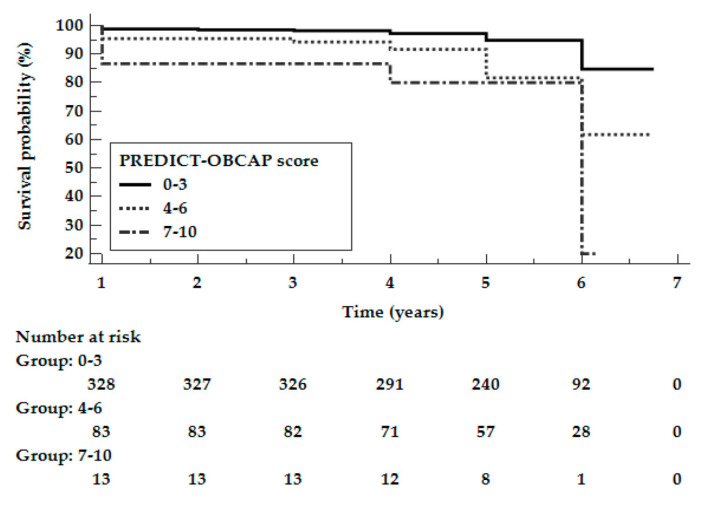
Kaplan–Meier curves according to the PREDICT-OPCAB score for the study population.

**Figure 3 jcm-10-03032-f003:**
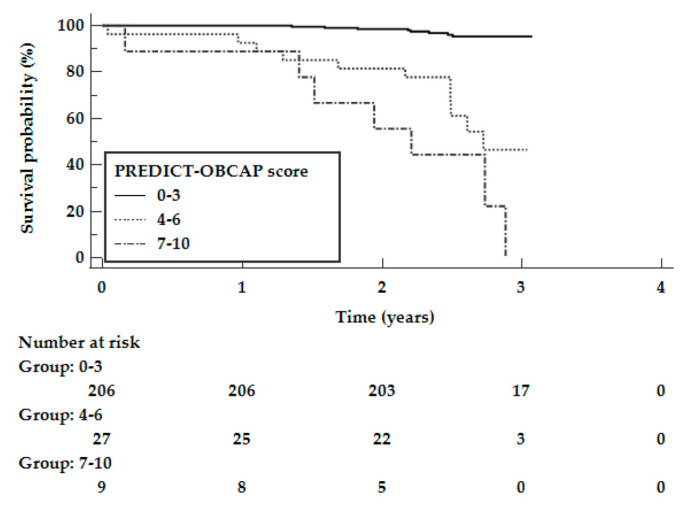
Kaplan–Meier curves according to the PREDICT-OPCAB score for the validation cohort.

**Table 1 jcm-10-03032-t001:** Clinical characteristics comparing patients with and without all-cause mortality endpoint during the median of 5.3 years follow-up.

	Survival *n* = 381 (87%)	Death *n* = 59 (13%)	*p*-Value
Demographic data			
Gender, male *n* (%)	332 (87%)	50 (85%)	0.8827
Age, years (mean ± SD)	64.1 ± 7.9	66.6 ± 8	0.0351
BMI, kg/m^2^ (mean ± SD)	28.5 ± 3.6	28.8 ± 4.2	0.4593
BSA, m^2^ (mean ± SD)	1.9 ± 0.2	1.9 ± 0.2	0.9404
Clinical data			
Left main disease *n* (%)	83 (22%)	18 (31%)	0.1276
Arterial hypertension *n* (%)	290 (76%)	49 (83%)	0.2349
DM *n* (%)	128 (34%)	22 (37%)	0.6517
Hypercholesterolemia *n* (%)	218 (57%)	24 (41%)	0.3829
COPD *n* (%)	34 (9%)	10 (17%)	<0.0001
Stroke *n* (%)	29 (8%)	10 (17%)	0.0261
PAD *n* (%)	53 (14%)	14 (24%)	0.0473
Chronic kidney disease *n* (%)	25 (7%)	8 (14%)	0.0642
Pre-operative laboratory parameters			
WBC, ×10^9^/L (mean ± SD)	8.0 ± 2.9	8.1 ± 2.0	0.4125
Neutrophils, ×10^9^/L (mean ± SD)	5.2 ± 2.3	5.6 ± 1.9	0.0715
Hb, mmol/L (mean ± SD)	8.7 ± 1.1	8.6 ± 0.9	0.1747
Plt ×10^9^/L (mean ± SD)	232.8 ± 63.0	239.1 ± 71.7	0.5670
NLR (mean ± SD)	3.2 ± 2.8	3.7 ± 3.6	0.0256
Pre-operative echocardiographic parameters			
LVEDD, mm (mean ± SD)	49 ± 6	51 ± 7	0.0049
LVEF, (mean ± SD)	54 ± 9	49 ± 11	0.0012
Intraoperative data			
Surgery time, h (mean ± SD)	2:39 ± 0:3	2:36 ± 0:4	0.4965
Anastomoses, number (mean ± SD)	2.3 ± 0.2	2.2 ± 0.2	0.0746
Blood product transfusion *n* (%)	151 (40%)	27 (46%)	0.3829
Length of stay, days (mean ± SD)	8 ± 3	8 ± 2	1.0000
Post-operative laboratory parameters			
WBC, ×10^9^/L (mean ± SD)	8.9 ± 3.2	9.1 ± 3.2	0.7186
Neutrophils, ×10^9^/L (mean ± SD)	5.4 ± 2.4	6 ± 2.9	0.2165
Hb, mmol/L (mean ± SD)	6.9 ± 0.8	7 ± 0.8	0.9804
Plt ×10^9^/L (mean ± SD)	287.1 ± 106.9	283.4 ± 98.2	0.5843
NLR (mean ± SD)	3.0 ± 1.9	3.8 ± 3.2	0.0173
Injury markers:			
Troponin T, ng/mL (mean ± SD)	4.33 ± 14.2	3.66 ± 5.9	0.3726
CK-MB mass, ng/mL (mean ± SD)	18.7 ± 33.4	19.3 ± 28.9	0.5293
Post-operative echocardiographic parameters			
LVEDD, mm (mean ± SD)	48 ± 7	51 ± 7	0.0017
LVEF, (mean ± SD)	55 ± 9	49 ± 11	0.0010

Abbreviations: BMI—body mass index, BSA—body surface area, COPD—chronic obstructive pulmonary disease, CK-MB—creatine kinase myocardial band, DM—diabetes mellitus, Hb—hemoglobin, LVEDD—left ventricle end-diastolic diameter, LVEF—left ventricle ejection fraction, NLR—neutrophil-to-lymphocyte ratio, PAD—peripheral artery disease, Plt—platelets, WBC—white blood count.

**Table 2 jcm-10-03032-t002:** Cox regression univariable analysis.

Parameter	HR	95% CI	*p*-Value
Baseline parameters			
Age	1.04	0.98–1.08	0.0531
Gender (male)	0.64	0.26–1.54	0.3200
Comorbidities			
Arterial hypertension	1.30	0.61–2.77	0.4838
Left main disease	1.98	1.12–3.49	0.0180
Diabetes mellitus	1.03	0.58–1.79	0.8962
COPD	2.07	1.43–4.11	0.0373
Hypercholesterolemia	0.61	0.35–1.04	0.0706
Chronic kidney disease	1.65	0.74–3.65	0.2150
Stroke history	2.40	1.21–4.76	0.0122
Pre-operative laboratory parameters			
WBC	1.01	0.94–1.09	0.6143
Lymphocytes	0.76	0.50–1.15	0.2061
Neutrophils	1.08	0.97–1.21	0.1253
Hemoglobin	0.86	0.67–1.12	0.2801
Platelets	1.00	0.99–1.00	0.3800
NLR	1.07	0.99–1.15	0.0714
Post-operative laboratory parameters			
WBC	1.02	0.96–1.08	0.4668
Lymphocytes	0.68	0.45–1.04	0.0827
Neutrophils	1.10	1.00–1.20	0.0323
Hemoglobin	1.05	0.75–1.48	0.7521
Platelets	0.99	0.99–1.00	0.6761
NLR	1.13	1.05–1.21	0.0011
Cardiac injury markers			
Troponins	0.99	0.96–1.02	0.8615
CK-MB mass	1.00	0.99–1.01	
Echocardiographic parameters			
LVEDD preoperative	1.05	1.01–1.09	0.0084
LVEDD postoperative	1.06	1.02–1.10	0.0031
LVEF preoperative	0.95	0.93–0.97	0.0002
LVEF postoperative	0.96	0.93–0.98	0.0015

Abbreviations: CKMB—creatine kinase myocardial band, COPD—chronic obstructive pulmonary disease, LVEDD—left ventricle end-diastolic diameter, LVEF—left ventricle ejection fraction, NLR—neutrophil-to-lymphocyte ratio, WBC—white blood count.

**Table 3 jcm-10-03032-t003:** Cox regression multivariable analysis.

Parameter	HR	95% CI	*p*-Value	Predict-Opcab Score
COPD	2.19	1.08–4.40	0.0278	2
Stroke history	2.67	1.34–5.31	0.0053	3
NLR after surgery ≥ 2.42	1.96	1.04–3.68	0.0364	2
LVEF after surgery ≤ 45	2.64	1.53–4.56	0.0005	3

Abbreviations: CI—confidence interval, COPD—chronic obstructive pulmonary disease, HR—hazard ratio, LVEF—left ventricle ejection fraction, NLR—neutrophil-to-lymphocyte ratio.

**Table 4 jcm-10-03032-t004:** Mortality rates in study population and validation population according to the PREDICT-OPCAB score.

Deaths	Predict-Opcab Score: 0–3	Predict-Opcab Score: 4–6	Predict-Opcab Score: 7–10	Log-Rank *p*-Value	Log-Rank *p*-Value 0–3 vs. 4–6	Log-Rank *p*-Value 0–3 vs. 7–10	Log-Rank *p*-Value 4–6 vs. 7–10
Study population *N* = 440	28 (8.3%)	24 (27.0%)	6 (40.0%)	<0.0001	<0.0001	<0.0001	0.1100
Validation cohort *N* = 242	8 (3.9%)	11 (40.7%)	7 (77.8%)	<0.0001	<0.0001	<0.0001	0.0838

**Table 5 jcm-10-03032-t005:** Clinical characteristics of patients with and without the all-cause mortality endpoint during the median of 2.6 years follow-up in the validation group.

	Survival *n*= 198 (89%)	Death *n*= 26 (11%)	*p*-Value
Demographic data			
Gender, male *n* (%)	174 (88%)	24 (92%)	0.5458
Age, years (mean ± SD)	65 ± 9	67 ± 9	0.2739
BMI, kg/m^2^ (mean ± SD)	22 ± 1.4	29 ± 5	0.4467
BSA, m^2^ (mean ± SD)	1.9 ± 0.2	1.9 ± 0.2	0.6726
Clinical data			
Left main disease *n* (%)	41 (21%)	5 (19%)	0.8132
Arterial hypertension *n* (%)	166 (84%)	21 (81%)	0.6921
DM *n* (%)	78 (39%)	9 (35%)	0.6385
Hypercholesterolemia *n* (%)	134 (68%)	18 (69%)	0.8733
COPD *n* (%)	11 (6%)	8 (70%)	<0.0001
Stroke *n* (%)	5 (3%)	10 (38%)	<0.0001
PAD *n* (%)	29 (15%)	11 (42%)	0.0005
Chronic kidney disease *n* (%)	18 (9%)	2 (9%)	0.7341
Pre-operative laboratory parameters			
WBC, ×10^9^/L (mean ± SD)	8.4 ± 3.3	7.7 ± 2.0	0.6769
Neutrophils, ×10^9^/L (mean ± SD)	5.3 ± 1.7	5 ± 1.5	0.7839
Hb, mmol/L (mean ± SD)	8.7 ± 0.9	8.7 ± 1.0	0.9882
Plt ×10^9^/L (mean ± SD)	229 ± 63	233 ± 62	0.9385
NLR (mean ± SD)	3.1 ± 1.8	3.2 ± 1.5	0.7109
Pre-operative echocardiographic parameters			
LVEDD, mm (mean ± SD)	47 ± 6	48 ± 6	0.5624
LVEF, (mean ± SD)	54 ± 8	50 ± 7	0.0030
Intraoperative data			
Surgery time, h (mean ± SD)	2:47 ±1:36	2:35 ± 1:26	0.5154
Anastomoses, number (mean ± SD)	2.3 ± 0.6	2.3 ± 0.7	0.9999
Length of stay, days (mean ± SD)	9 ± 3	13 ± 10	0.0030
Post-operative laboratory parameters			
WBC, ×10^9^/L (mean ± SD)	9.1 ± 4.9	12 ± 13	0.0330
Neutrophils, ×10^9^/L (mean ± SD)	5.1± 2.0	8.7 ± 11.6	0.0012
Hb, mmol/L (mean ± SD)	6.9 ± 0.6	7.0 ± 0.5	0.3563
Plt ×10^9^/L (mean ± SD)	305 ± 92	354 ± 107	0.0157
NLR (mean ± SD)	2.8 ± 1.5	5.1 ± 3.6	0.0003
Injury markers:			
Troponin T, ng/mL (mean ± SD)	3.9 ± 6	9.9 ± 11	0.0113
Post-operative echocardiographic parameters			
LVEDD, mm (mean ± SD)	47 ± 6	50 ± 6	0.0205
LVEF, (mean ± SD)	56 ± 8	43 ± 7	<0.0001

Abbreviations: BMI—body mass index, BSA—body surface area, COPD—chronic obstructive pulmonary disease, CK-MB—creatine kinase myocardial band, DM—diabetes mellitus, Hb—hemoglobin, LVEDD—left ventricle end-diastolic diameter, LVEF—left ventricle ejection fraction, NLR—neutrophil-to-lymphocyte ratio, PAD—peripheral artery disease, Plt—platelets, WBC—white blood count.

## Data Availability

Source data files are available upon request.

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
