# Peer review of "A Risk Score for Predicting Long-Term Mortality Following Off-Pump Coronary Artery Bypass Grafting"

_jcm, 2021, doi:10.3390/jcm10143032_

Round 1
Reviewer 1 Report
Authors have performed the creation of a short new score to predict long-term mortality after off-pump CABG.
The study is interesting, but I have several consideration about the manuscript.
Introduction:
.- I think is too large. An introduction must be short and introduce directly to the problem and the study objective (why authors have performed the study).
.- Line 67: “we hypothesized that the risk factors associated with adverse prognosis might differ as well”. If you affirm that the risk factor could be different, it`s mandatory to have a comparative group undergoing on-pump CABG.
Patients and Methods:
Study population:
.- Authors must clarify if the study is prospective or retrospective.
.- One of the exclusion criteria was the acute myocardial infarction. Does this criteria include NSTEMI or only STEMI?. If authors excluded all patients with acute myocardial infarction, that point must be explain in discussion, limitations and conclusions, and explain correctly that this new score is only validated in stable patients but not in patients with acute myocardial infarction.
.- Line 105-107: Phrase “under physiologic stress……than either alone” should be eliminated. It`s explained in introduction and must not be repeated in Methods.
Outcomes:
.- The outcomes are not according to results. Authors must add all the analysis performed as endpoints (risk factors analysis, score, score validation,…).
Results:
Predictive value of NLR for long-term mortality
I think is not methodologically correct not include those variants with a p value close to significance like age. When authors using the score to validate it, the mean follow-up is very short regarding those cohort where the score has been validated.
Discussion:
Must be started with a summary of the main findings of the study. The current discussion is large and not clear at the beginning. And I think the main aspect of the study is not the NLR. The most important result of the study is a new score that predicts the long-term mortality in patients undergoing OPCABG including the NLR as a significant risk factor.
Conclusions:
For me the main finding of the study is the score for the prediction of mortality at long-term follow-up after OPCABG. For these reason, I think conclusions should be change. The main aspect of the study is the new score, not the NLR value and their inclusion in the score.
Author Response
Dear Reviewer,
we are thankful for the time and effort that you spent to provide in-depth review of our manuscript. We corrected our manuscript according to your suggestions. Our response is presented below. All changes in the new version are marked in red color in manuscript. The answers to your questions that not appeared in text are marked in yellow color.
On behalf of all Authors,
Sincerely,
Tomasz Urbanowicz, MD, PhD

Reviewer 2 Report
Summary
This is a retrospective study aiming at analyzing risk factors for survival after off-pump coronary artery bypass grafting. The authors report a variety of clinical and laboratory variables associated with late mortality, and suggest a risk score to predict mortality.
Comments:
Comment 1
L49-50
The term ‚long-term’ is unspecific. Please give the failure rate in relation to the time after surgery, e.g. 10-year patency rate of grafts.
Comment 2
L114-124
Is OPCAB the standard approach for CABG at your institution? When do you perform on-pump CABG, and when OPCAB? Please explain.
Comment 3
L114-124
Did all patients undergo surgery with LIMA and GSV ? Did some patients receive LIMA+RIMA? Did some patients receive only venous grafts? This is important to know as it could have an impact on long-term survival.
Comment 4
L114-124
Were there any conversions from OPCAB to on-pump surgery?
Comment 5
L176-177
In my opinion, some important variables are missing. I suggest expanding the variables in order to get a more precise risk factor analysis. E.g. previous cardiac surgery, postoperative respiratory insufficiency, stay on ICU, perioperative stroke, rethoracotomy for bleeding, rethoracotomy for bypass revision, postoperative coronary angiography, etc.
Author Response

(The authors gave the same response as above.)

Round 2
Reviewer 1 Report
Authors have performed correctly the changes of the manuscript that I suggested them. The quality of the manuscript has clearly improved after the reviewers suggestions.
Author Response
Dear Reviewer,
thank you for you time and valuable suggestions, you have given to us.
Kind Regards
Tomasz Urbanowicz & co-authors
Reviewer 2 Report
Thank you for your revision.
Author Response
Dear Reviewer,
thank you for your time and effort that allowed us improve our manuscript.
We have performed spell check according to your suggestions including:
line 133 - word corrected: "disease"
linÄ™ 143 & 256 - word corrected "endpoint"
line 384 - word corrected "performed"
Kind regards
Tomasz Urbanowicz & co-authors
This manuscript is a resubmission of an earlier submission. The following is a list of the peer review reports and author responses from that submission.
Round 1
Reviewer 1 Report
Thank you very much for your paper,
in my opinion you have to highlight the need for a new risk score in this patints' setting.
good luck for the paper.
Author Response
Dear Reviewer,
we are thankful for the time and effort that you spent to provide in-depth review of our manuscript.
On behalf of all Authors,
Sincerely,
Tomasz Urbanowicz, MD, PhD
Michał Michalak, MD, PhD, Assoc. Prof.
Aleksandra Gasecka, MD, PhD
Reviewer 2 Report
Urbanowicz et al. developed a novel risk score that predicts long-term mortality after off-pump coronary artery bypass graft surgery for MVD in a Polish single-center registry. Their study population included 440 consecutive patients with a median follow-up of 5.3-years. The authors developed a risk prediction model including the neutrophil to lymphocyte ratio; an easy to obtain inflammatory marker. They found that the NLR was a significant risk predictor for long-term mortality alongside history of stroke, COPD, and LVEF. I find the author's attempts thought-provoking yet its clinical significance remains questionable. I would appreciate if the authors can elaborate or correct the following comments that I have made.
Major comments
- I agree that there isn’t a risk score predicting mortality after OPCAB. However, is a model solely applicable for MVD OPCAB useful or actionable in anyway? What are the clinical implications for using this model to risk stratify patients? Is there anything that can be done prior to/during or after surgery to alter the course of MVD OPCAB patients? This should be clearly stated in the Introduction to highlight the importance of this study.
- Why was the primary outcome chosen to be mortality? It would be more practical if it were to predict cardiac deaths, nonfatal MI, stroke and graft failure since all-cause mortality may only reflect non-cardiac deaths associated with the comorbid state of patients undergoing OPCAB.
- Surgical outcomes may differ greatly vary by operator experience. Were the 440 operations performed by the surgical team or chief operator? Is there a cutoff value of number of surgeries that determines the quality of surgery in Poland?
- Why was NLR sampled on -1POD and 7POD? The authors should describe why they chose these timepoints? Please give reference from previous studies if any.
- Selection of covariates in a forward/backward selection are generally no recommended nowadays as compared to selecting clinical important variables based on studies from a similar field (eg. EuroScore, STS-SCORE).
- How was model discrimination and calibration evaluated in this study? Please give specific statistical methods that the authors chose in the Methods section.
- When were the cardiac injury markers sampled? Please clarify.
- The authors should also generate and validate the performance of a risk model containing values that are continuous such as LVEF and NLR. Although to create bedside risk score, creating a cutoff value and a dichotomous value may be easier, this also leads to lessening the performance of a predictive model. Please also demonstrate a histogram for continuous variables to observe their distribution (parametric or non-parametric).
- I find it troubling to use a LVEF measured after the surgery, when in case the risk model is intended to give predictions prior to the surgery. Would it be possible to replace this with a LVEF measured prior to the operation?
- The population used to create the KM curve in Figure 1 seems to lack number of events. Also, please give a number at risk table on the bottom of the figure and retitle the x title with proper units (eg. Years? Months?)  
- The authors may reconsider to reform their message by simply stating that NLR was a predictive factor for all-cause mortality among OPCAB patients and not go to deeply into creating models or validating models since the study population and number of events seem to be lacking.
Minor comments
- Delete arterial from Line 130.
- chronic obstructive disease should be chronic obstructive pulmonary disease in Line 131.
- Please give mortality rates at 1-year in 2 Risk factors of mortality and development of the risk score €i0
- P values should generally be described in no more than two decimal points (eg p=0.0351 should be p=0.04) in the Results Section, text and Tables.
- Line 160; Please give p values after HR=0.95, 95% CI 0.93 - 0.97 and HR=0.96, 95% CI 0.93 - 0.98 in Line 160.
- Table 2; the HR values in the table should be correctly moved to the corresponding parameters, respectively.
Author Response
Dear Reviewer,
we are thankful for the time and effort that you spent to provide in-depth review of our manuscript. We corrected our manuscript according to your suggestions. Our responses to your suggestions are presented in the attached file.
On behalf of all Authors,
Sincerely,
Tomasz Urbanowicz, MD, PhD
Michał Michalak, MD, PhD, Assoc. Prof.
Aleksandra Gasecka, MD, PhD

Reviewer 3 Report
The manuscript deals with the topic of off-pump coronary artery bypasses (OPCABs), which represent 15-30% of the total CABG. In particular the authors finalize the study to create a specific risk score for long-term mortality for this type of revascularization procedure. The study is well designed, the only criticism is in the sample size which could have been more numerous, although the statistical analysis is correct, an independent population was considered as a validation cohort. The primary outcome was long-term all-cause mortality after OPCAB, the secondary outcome was the pre- and postoperative predictive value of Neutrophil to lymphocyte ratio (NFL) as an indicator of inflammatory response. The multivariate Cox regression model identifies four independent predictors of mortality after OPCAB. After weighing these variables in the multivariate regression analysis, the authors developed a specific risk score model for OPCAB called PREDICT-OPCAB Score. We congratulate the authors on how they structured the study and on the creation of a specific risk score for OPBCAB. I invite the authors to propose this risk score model in multicenter studies in order to further validate it with a larger patient population.
Author Response
Dear Reviewer,
we are thankful for the time and effort that you spent to provide in-depth review of our manuscript.
We would be delighted to have a chance to launch a multicenter study based on our predict-score with you.
I allow myself to send you contact email: tomasz.urbanowicz@skpp.edu.pl.
On behalf of all Authors,
Sincerely,
Tomasz Urbanowicz, MD, PhD
Michał Michalak, MD, PhD, Assoc. Prof.
Aleksandra Gasecka, MD, PhD
Reviewer 4 Report
Thank you for the opportunity to review this manuscript written by Dr. Urbanowicz and colleagues. In this study, they have reviewed 440 patients undergoing isolated off-pump CABG to develop a risk score to predict long-term mortality. In the model, they include traditional/classical variables such as patient demographics and comorbidities, but they also investigate the influence of neutrophil:lympocyte ratio (NLR) which is a marker of inflammation shown to have predictive value in other cardiovascular arenas. They then validated the score in an independent sample of 242 patients. The model appears to have fair discrimination among the low risk group versus moderate/high risk groups, though less so among the moderate and high risk cohorts. I have several comments/questions for the authors.
Major:
- The authors are correct that there are limited options available to prognosticate long-term survival after cardiac surgery, in particular OPCAB. There is also interest around "non-classical" risk factors, such as inflammatory markers. However, I think the authors are trying to combine two studies into one in this report. There is a lot of information to follow for a reader. The paper may benefit from separating into two questions (ie, two reports): first, what is the influence of NLR on outcomes in general--short term and long term mortality, as well as morbidity. Second, once NLR is fairly established as a legitimate prognostic factor, a full model for long-term mortality. As it stands in the current report, readers are asked to buy into the concept of NLR as a predictor with limited understanding and evidence that it matters.
2. The concept of NLR needs better explanation/context in the introduction and discussion. The authors do provide several helpful citations, but readers will need more context in this report. How is it calculated exactly? What factors influence whether the value is high or low? Why do the authors think the value influences (or is associated with) outcomes? This information may need to be dispersed where appropriate to the Intro, Methods, and Discussion.
3. Unfortunately, including post-operative NLR in the model is not very helpful because, de facto, this value cannot be known preoperatively. Why did the authors elect to include this in the model? The model should only include variables that are known pre-operatively, otherwise there is no ability to prognosticate without guessing or hoping about some value that will not be known until 1 week after surgery. If the authors insist on keeping postoperative NLR in the model, they should provide further justification for this decision. As above, it seems they are forcing NLR into the model in any way they can, and instead should separate out these questions into two studies.
4. To further support that NLR matters, it would be interesting to compare NLR values from baseline to postop in ONCAB and OPCAB patients. I realize that is not the focus of this particular report, but if NLR were to change differently between the techniques, this would lend further credence to study of NLR in the OPCAB population.
5. While the risk-factors identified may prognosticate long-term outcomes, none seem to be modifiable except perhaps NLR, and this is not clear. The authors should list this as a limitation. They should also discuss whether NLR is modifiable, and if so how? Are the authors taking measures to "optimize" NLR in their patients prior to OPCAB? How so?
6. Again, NLR seems to be getting forced into the model. Its predictive performance is relatively mediocre with an AUC of 0.6. It is only slightly better than a coin flip. More justification is needed as to why this variable should be included in the model.
Minor:
- Abstract: the validated performance of the model (AUC) should be reported
- Introduction: The authors cite two examples of risk models for short and long term mortality after CABG. They should also include a reference to the STS score and EuroScore.
- Methods: please state precisely how NLR is calculated
- Results: towards the end, things seem to get out of order. In section 3.3, a Table 5 is presented on the validation cohort, but the text does not catch up until section 3.4. Also, validation results are reported in Figure 2, which comes before many other tables/figures that need to be presented first.
- Results: no in-text reference to Figure 2 is present
- Discussion: the authors need to provide further discussion on the points listed above.
Author Response
Dear Reviewer,
we are thankful for the time and effort that you spent to provide in-depth review of our manuscript.
We've enclosed the responses to your valuable suggestions below.
On behalf of all Authors,
Sincerely,
Tomasz Urbanowicz, MD, PhD
Michał Michalak, MD, PhD, Assoc. Prof.
Aleksandra Gasecka, MD, PhD

Round 2
Reviewer 1 Report
Thank you very much and good luck for your paper
Reviewer 2 Report
I believe that the authors have put much effort into answering my comments sent previously. Only several spelling errors have been noted within the text that should be corrected prior to publication.
Line 16 and 54 Spell out STS since it is an abbreviation
Line 34
All-cause mortality after OPCAB is influenced both >>>
All-cause mortality after OPCAB was influenced both
Reviewer 4 Report
Thank you to the authors for their revisions and response to the reviewer comments. I remain concerned regarding the use of postoperative NLR as a prognostic tool. Predictive models are really only clinically useful in the preoperative phase of care, and therefore any predictive model should include only information available in the preoperative phase. The authors argue that use of their model may identify patients at risk postoperatively that may benefit from this awareness of their increased risk--but offer no solutions about how to care for these patients to improve outcomes other than to intensify follow-up care in a vague way.
I do not think the risk score the authors are presenting is clinically useful. The authors have collected a large series of OPCAB patients with thorough data. At this point, I would be more interested if they used the data to create a model without the postoperative NLR value and using only preoperative variables. The low predictive value of NLR on its own makes it a curious choice for force into a multivariate model, in any case. If the authors insist on studying NLR, they should do so independent of a prognostic model and instead focus on what interventions would improve NLR postoperatively or prevent NLR from increasing after surgery.